# Towards geospatially-resolved public-health surveillance via wastewater sequencing

Braden T. Tierney [1,2] ✉, Jonathan Foox[1], Krista A. Ryon[1], Daniel Butler[1], Namita Damle[1], Benjamin G. Young[1], Christopher Mozsary[1], Kristina M. Babler [3,4], Xue Yin[4], Yamina Carattini[5], David Andrews[5], Alexander G. Lucaci[1,2], Natasha Schaefer Solle[6], Naresh Kumar[7], Bhavarth Shukla [6], Dušica Vidović[8,9], Benjamin Currall[9], Sion L. Williams[9], Stephan C. Schürer [8,9,10], Mario Stevenson[6], Ayaaz Amirali[4], Cynthia Campos Beaver [6,9], Erin Kobetz[6,9], Melinda M. Boone[9], Brian Reding[11], Jennifer Laine[11], Samuel Comerford[11], Walter E. Lamar[12], John J. Tallon Jr[13], Jeremy Wain Hirschberg[1], Jacqueline Proszynski[1], Gabriel Al Ghalith[14], Kübra Can Kurt [1], Mark E. Sharkey[5], George M. Church [15], George S. Grills[9], Helena M. Solo-Gabriele [4] ✉ & Christopher E. Mason [1,2,16] ✉

Wastewater is a geospatially- and temporally-linked microbial fingerprint of a given population, making it a potentially valuable tool for tracking public health across locales and time. Here, we integrate targeted and bulk RNA sequencing (N = 2238 samples) to track the viral, bacterial, and functional content over geospatially distinct areas within Miami Dade County, USA, from 2020-2022. We used targeted amplicon sequencing to track diverse SARS-CoV-2 variants across space and time, and we found a tight correspondence with positive PCR tests from University students and Miami-Dade hospital patients. Additionally, in bulk metatranscriptomic data, we demonstrate that the bacterial content of different wastewater sampling locations serving small population sizes can be used to detect putative, host-derived microorganisms that themselves have known associations with human health and diet. We also detect multiple enteric pathogens (e.g., *Norovirus*) and characterize viral diversity across sites. Moreover, we observed an enrichment of antimicrobial resistance genes (ARGs) in hospital wastewater; antibiotic-specific ARGs correlated to total prescriptions of those same antibiotics (e.g Ampicillin, Gentamicin). Overall, this effort lays the groundwork for systematic characterization of wastewater that can potentially influence public health decision-making.

Wastewater is a wellspring for geospatially delineated, epidemiologically relevant public health data. One sample contains a cross-sectional collection of human pathogens, commensals, animal/plant detritus, biomolecules, and environmental features (e.g., pollutants) deriving from a specific locale[1]. Depending on the sampling location, it represents small populations, like the residents of one building or entire cities[2]. As a result, wastewater is a potential source for continuous, precise monitoring of public health and potential pathogens for communities[3]. Applications of wastewater-based epidemiological surveillance include tracking community drug abuse, pollutants, and pathogen load[4-6].

The SARS-CoV-2 pandemic galvanized efforts to develop low-cost tools for pathogen surveillance. Targeted sequencing of SARS-CoV-2 wastewater gained popularity for its advantages over clinical or individual testing-based approaches[7]. Wastewater surveillance can (1) predict outbreaks before cases spike[8], (2) eliminate individual patient level data, reducing privacy concerns, (3) capture undersampled populations that may not be regularly tested[9], (4) be more cost-effective than doing repeated individual tests[10], and (5) reduce the reliance on self-reported data, which has proven a challenge in estimating both global SARS-CoV-2 burden and individual vaccination status[11]. Numerous prior studies have explored the use of targeted RNA sequencing in tracking pathogens, most notably SARS-CoV-2. These have been executed in numerous areas around the globe, including the US, Germany, Switzerland, Japan, Hong Kong, India, and Africa[12–19].

However, the utility of wastewater surveillance extends beyond monitoring single pathogens. Many past wastewater studies tracked environmental pollutants, like heavy metals and fertilizer runoff[20]., or sewage in canals[21]. Recent efforts have used a combination of targeted (e.g. Polymerase-Chain-Reaction(PCR)-based) and untargeted (e.g. shotgun) amplification approaches to characterize multiple pathogen abundances simultaneously, including clinically relevant enteric viruses[22]. These have revealed uncharacterized microbial life in wastewater, some of which have the potential to be biomarkers for aspects of human and societal health, ranging from nutrition (derived from metabolomics and 16S ribosomal RNA sequencing) to cardiovascular disease and cancer[23,24]. More specifically, prior efforts have proved successful at the detection of and culturing of bacterial pathogens and antimicrobial organisms in wastewater[24–28].

To leverage these advances and explore the potential of microbiome sequencing in monitoring public health, we collected and amplicon-sequenced 966 (weekly, on average) samples from 34 sites in Miami-Dade County between 2020 and 2022 (Fig. 1a, Supplementary Data 1). We performed targeted qPCR and targeted sequencing to estimate total SARS-CoV-2 and variant proportions. On an additional set of samples, ($N = 1272$), we completed bulk RNA sequencing to ascertain the broader microbial community. We sampled five diverse location types: primary/secondary schools (median population size of 1003 individuals across nine distinct sites), the University of Miami Campus wastewater basin (median population size of 1088 individuals across six distinct sites), dormitories (median population size of 370 individuals across 16 distinct sites), hospitals (median population size of 340 individuals across five distinct sites), and a regional wastewater treatment plant (representing a population of over 800,000). We developed standardized collection and analysis workflows to holistically evaluate the microbial content of wastewater and generate public-health-relevant guidance (Fig. 1a)[7,29–34].

Bioinformatically, the analysis process involved quality control (i.e., filtering targeted sequencing data for samples with >75% SARS-CoV-2 genome coverage at >10×) as well as, in the case of bulk data, processing with multiple algorithmic workflows (i.e., kmer-based taxonomic classification, alignment-based taxonomy, de novo assembly-based approaches) and databases to yield multiple taxonomic and functional calls for each sample. The end result is a report integrating multiple views into the composition of each wastewater sample during various waves of the COVID-19 pandemic.

## Results

### Geospatially resolved wastewater vs. clinical SARS-CoV-2 variant tracking

We used targeted sequencing to generate high-resolution maps of SARS-CoV-2 Variant of Concern (VoC) levels across time based on coverage in wastewater sequencing data (Supplementary Data 1). Our sequencing and analytic workflow was executed in part to reproduce other substantial efforts[8,14,35] to monitor SARS-CoV-2 VoC levels in wastewater while also adding the dimension of geospatial and population size variation in sampling locations. These past efforts have shown that wastewater VoC abundance correlates with population VoC abundance in clinical samples and can predict outbreaks[15,36]. We hypothesized that different sites provided distinct levels of lead time, for example, in identifying outbreaks before clinical testing. In total, we observed across all wastewater samples the rise and fall of the VoCs that dominated the pandemic up to the Omicron BA5 strain. Wastewater VoC abundance was positively correlated with VOCs (Patient vs. Wastewater: rho = 0.53, $p < 0.001$, Student vs. Wastewater: rho = 0.61, $p < 0.001$) in 5442 sequenced or otherwise characterized samples from University of Miami (UM) students ($N = 1503$) and the UM UHealth hospital patients ($N = 3939$ patients) (Fig. 1b, c).

The initial observation of different VoCs in wastewater was coincident with their initial identification in these other samples, especially data later in the pandemic as dominant variants became more defined. We additionally observed that the monthly proportion of VoCs in wastewater varied over time (Fig. 1b, c), especially compared to the proportion of VoCs detected in the clinical and student settings (Fig. 1b); in other words, wastewater VoCs appeared to track the monthly waves the pandemic more effectively that individual testing. Variant detection in wastewater preceded either (or both) student/patient detection for 5/8 VoCs identified in 2021 and 2022 (Supplementary Data 1). On average, VoCs in wastewater were identified within 1.8 days of their identification in either student or patient samples. Notably, the delta variant was detected eight days in wastewater prior to its detection in student samples.

Not only did the wastewater data provide an antecedent view of circulating strains in patients, the data also gave a more complete picture of all variants circulating in Miami-Dade County than either the student or patient data alone, especially since these approaches relied on random sampling-based with single individuals being tested (Fig. 1c). For example, the Mu variant was identified in the wastewater and patient data (at high coverage for the clade-specific variants), but not in the student data, further indicating the propensity for individual testing to overlook potential circulating VoCs.

In addition, the continual sampling of different wastewater sites enabled us to observe spatial variation in variant detection within the different sites we collected wastewater (Fig. 1d). For example, the Eta variant was present in March/April 2021 in the UM Campus Basin, whereas Lambda was more detectable in wastewater treatment plants and UM dormitory sites. Mu was observed in hospital wastewater but in no other wastewater collection site. In this sense, geospatially resolved, continual targeted monitoring provided transmission information for alternative variants from potentially survey-undersampled, non-hospital-associated communities for alternative variant transmission.

As expected, wastewater-based VoC surveillance enabled tracking of the mutational-level transitions between dominant strains across time (Fig. 2). We were able to observe transitions in both unique and recurrent mutations between VoCs. This high-resolution phylogenetic tracking enabled monitoring of the rise and fall of competing variants across space. We attained clear tracking of the variant mutational landscape despite using only weekly sampling.

### Wastewater contains human gut microbial taxa that have reported associations with nutrition and host health

While the ability of targeted sequencing to track specific, low-abundance pathogen load in wastewater has been demonstrated, the potential utility of bulk RNA sequencing for monitoring public health has been explored in only a few instances, predominantly for estimating virome composition[37]. We hypothesized, however, that the bacterial gene expression of wastewater could provide signals correlated to either environmental health (e.g., correlation to contaminant abundance) and/or bioindicators for human community health. To provide an initial basis of evidence for this hypothesis, we searched for

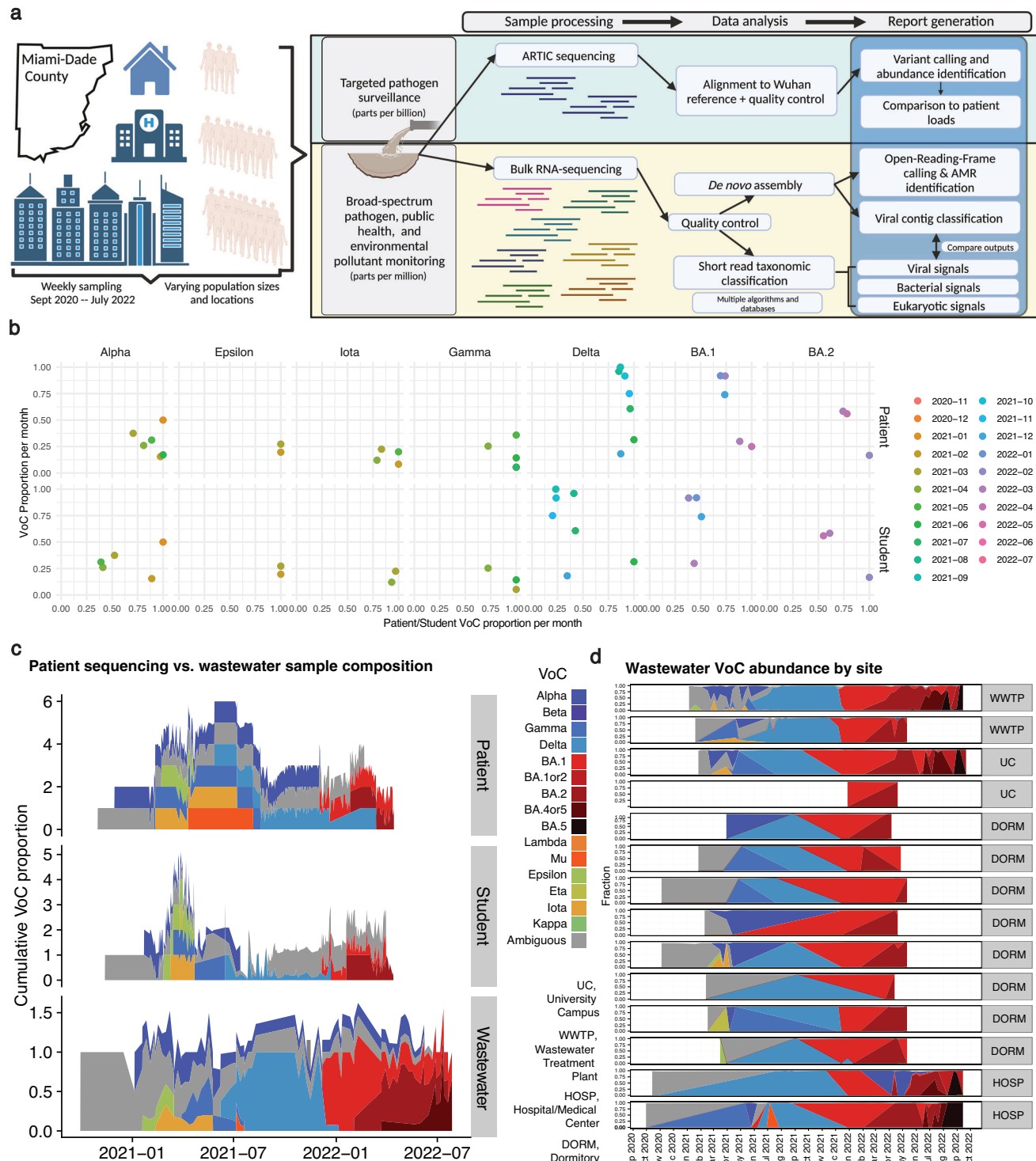

**Fig. 1 | Overview of approach and targeted sequencing of SARS-CoV-2.**
**a** Samples were taken at (on average) weekly intervals between 2020 and 2022 from 34 sites within Miami-Dade County. Samples were sequenced with targeted ARTIC sequencing to measure SARS-CoV-2 abundance and bulk RNA sequencing to ascertain the broader microbial community. A variety of algorithms and analytic approaches were employed to identify and compare the taxonomic and functional profiles of each site across time and space, with the end result being a systematically characterized dataset with comparisons to clinical data to provide information relevant to public health surveillance showing the evolution of variants over space and time. **b** The monthly VoC proportions across datasets. Point color

corresponds to the month. X-axis is the proportion of samples annotated as a given VoC for patient or student samples (derived from individual tests). Y-axis is the average variant abundance in targeted wastewater sequencing. **c** An additional, density-plot-based, view of all variants in wastewater vs patient/student cohorts over time. Colors correspond to different variants as defined in the legend between (**c**) and (**d**); this legend is relevant to both panels. **d** The variation in wastewater VoCs across time in different sampling sites. Source data are provided as a Source Data file. **a** created with BioRender.com released under a CC-BY-NC-ND 4.0 International license.

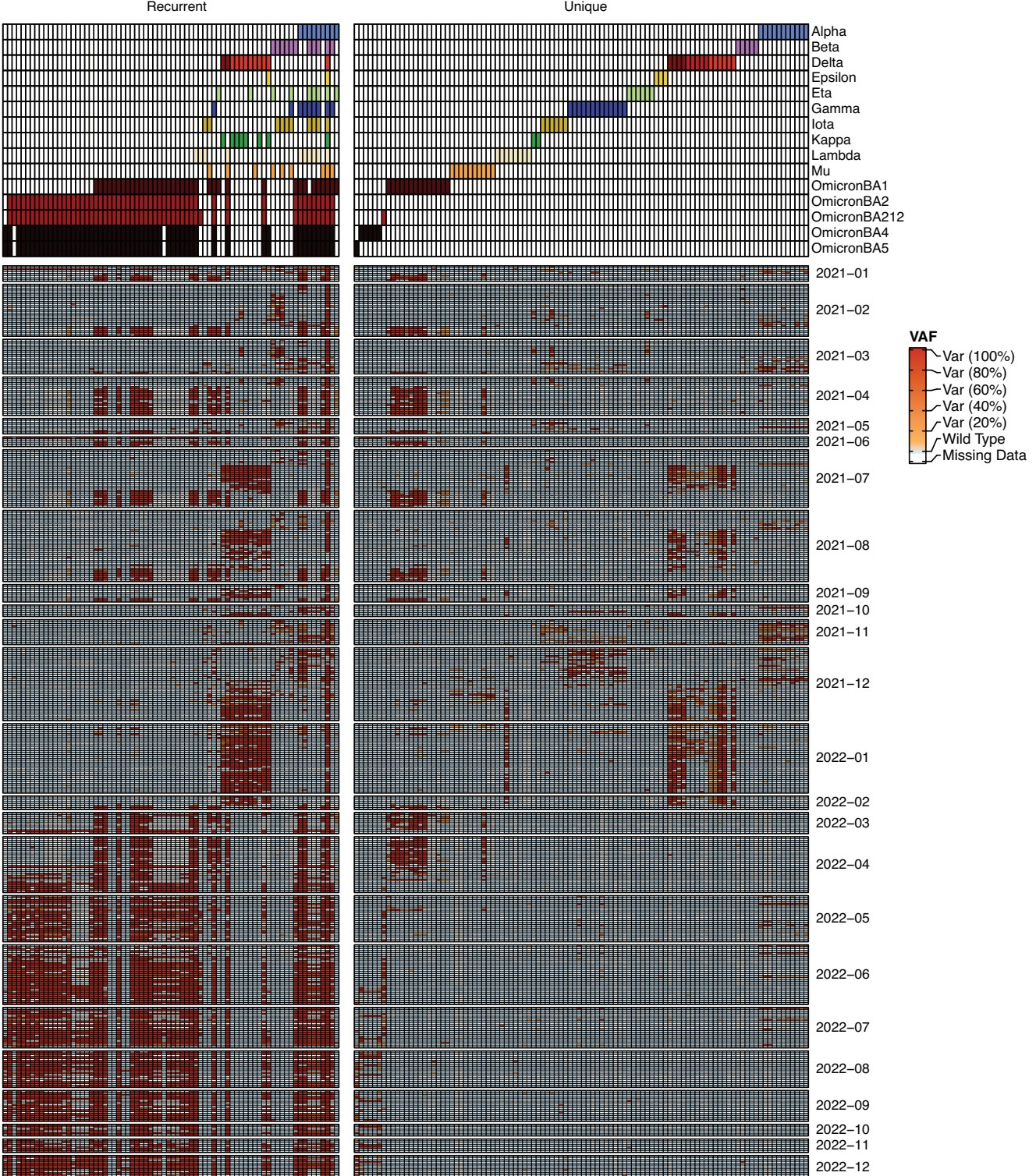

**Fig. 2 | Tracking Variants of Concern (VOCs) across time.** For every wastewater sample, fragments deriving from SARS-CoV-2 were analyzed for mutations corresponding to COVID variants of concern (VOC), and the Variant Allele Frequency (VAF) was estimated per mutation per sample. Unique mutations refer to point mutations that are unique to a given VOC, while recurrent mutations are present in more than one VOC. The annotation track on top reflects which VOC corresponds to which mutation (columns). This annotation track has unique colors in each line, and the colors are present only to assist in delineating between each subsequent row. Each cell in the dated rows represents the VAF for a given mutation for a given sample at a given time point, with dark red reflecting all RNA fragments having a given mutation, lighter red reflecting a mixture of presence and absence for a mutation, blue representing the presence of the wild type (i.e. not deriving from a given VOC), and gray representing no coverage of that genomic context in the sequence data. Source data are provided as a Source Data file.

microbial signals – both pathogenic and commensal – in different sites that were derived clearly from taxa known to be associated with variation in human nutrition or gut health.

We collected 1272 bulk RNA sequencing samples. After read quality control, these data comprised 14.7 billion read pairs at a median per sample sequencing depth of 11,802,903 reads. This included control samples, which we used to bioinformatically filter potential processing and sequencing contaminants (see "Methods" section). We report the taxa removed by this contaminant filtering process in Supplementary Data 2. We compared multiple taxonomic annotation approaches and found complementary results in the profiles they produced (Supplementary Fig. 1). We recorded metadata on all samples and assemblies, and computed mapping summary statistics (Supplementary Data 3). With Kraken2, only a small percentage (median = 3.71%; first percentile = 1.80%, 99th percentile = 16.25%) of our reads went unclassified at the Domain level. While concordant in terms of genera identified, however, the three taxonomic classifiers we implemented identified different numbers of species, with Kraken2 capturing the most, followed by Xtree, followed by MetaPhlan4. While this could derive from variation in database size and/or sensitivity, we hypothesize the substantially fewer taxa identified by MetaPhlAn4 could arise from its classification strategy stemming from marker genes instead of whole-genome based kmer-matching; metatranscriptomics will only capture transcriptionally active portions of a genome, and there is no reason why active genes would be those that are discriminating in MetaPhlAn4's database. Alternatively, kmer-based, whole-genome approaches could provide inflated species estimates due to false positive alignments. Xtree had the most overlap with both classifiers, in total, and as such, for the rest of this section, we report in figures the Xtree classifications that were found additionally by at least one other taxonomic classifier, unless otherwise stated.

We first observed that, as to be expected, the bacterial content of wastewater comprised an amalgamation of both host-associated clades as well as microbes normally found in the natural environment (Fig. 3a). We detected many human gut commensal families (e.g. *Ruminococcaceae, Lachnospiraceae, Akkermansiaceae, Bifidobacteriaceae*) as well as oral microbiome commensals (e.g., *Nanogingivalaceae, Fusobacteriaceae*). Outside of host-associated organisms, many different environmental families, including those previously reported in wastewater (e.g. *Rhodocyclaceae, Nitrospiraceae*) were also present[38,39]. The average abundance of these physiologically diverse (i.e., spanning both anaerobes and aerobes) taxa varied as one would expect; clades more abundant in the wastewater treatment plant were previously found to also be prevalent in activated sludge[40], whereas the human-associated organisms were more abundant in other sites. Overall, the amalgam of human and environmental microbes increased our confidence that it could potentially contain a broad spectrum of biomarkers indicative of different public health-relevant indications.

To better understand the patterns of site-specific bacterial abundance variation observed in Fig. 3a, we next interrogated the seasonal variation in bacterial microbiome content as a function of sampling location and population size (Fig. 3b). Measuring within-site variation (i.e., beta diversity, the dissimilarity between any pair of samples from a given site), we found that each sampled location had distinct temporal signatures. Specifically, there was an inverse relationship between population size and average within-site variation. Areas serving low populations – most notably, dormitories, demonstrated high variation in wastewater bacterial content between any two samples; conversely, the wastewater treatment plant samples had low variation and served a massive population.

We next aimed to identify the source of this site-specific difference in variation, hypothesizing that it likely arose from high variation in some of the site-specific families present in Fig. 3a. We computed overlap in identified bacterial species between each site as well as each species average abundance and prevalence within a given location

type (dormitories, the entire university campus, primary/secondary schools, wastewater treatment plants, hospital/medical campus buildings) (Fig. 3c). A total of 182 species were present in at least one sample from each of these five location types. These comprised predominantly members of the wastewater-specific and/or environmental families (Fig. 3a). It was clear, however, that the sample-to-sample variation in dormitory microbiota, as observed in Fig. 3b, was driven by a large number (N = 646) of unique species that were both (1) high abundance (Fig. 3c, middle) and (2) low (i.e., less than 10%) prevalence (Fig. 3c, top). The overlaps between other sites were as expected; for example, the University of Miami locations all shared more taxa with each other than with the wastewater plant or the primary/secondary schools.

To compute the microbiome architecture of wastewater[41,42], we completed a Microbial Association Study (MAS) between species abundance and sampling sites, representative population size, and various environmental features, like salinity and pH (Supplementary Fig. 2). We considered a significant association as having a Benjamini–Yekutieli (BY) p-value below 0.05. Bacterial composition was associated most strongly with location type and population size, as expected, and these results were reproduced across taxonomic classifiers (Supplementary Data 4). We confirmed that the organisms associated with dormitories and other sites with low populations tended to be known human gut commensals (Fig. 3d, Supplementary Data 4) and potentially pathogenic microbes (e.g., *Klebsiella pneumoniae*).

Overall, in wastewater samples representing small populations (<1000 individuals) the bacterial content comprised numerous taxa typically observed in the gut microbiome that are known to be associated with dietary intake For example, *F. prausnitzii, R. hominis, Anaerostipes hadrus, Agathobaculum butyriciproducens*, and *B. animalis*, all found in our wastewater data, have been associated with healthy plant-based and animal-based food intake[43]. Similarly, other organisms found (e.g., *Clostridium leptum, Bifidobacterium bifidum, Eggerthella lenta, Ruthenibacterium lactatiformans, Clostridium innocuum*), have been reported as associated with poor diets[43]. This indicates that geospatially variable wastewater may be useful for tracking population dietary choices in wastewater samples representing small (<1000 individuals, as in the dormitories in this study) populations.

Moreover, many of these small-population-associated species are also potential gut-based indicators of human disease or health status. For example, *F. prausnitzii* has been repeatedly and robustly reported as inversely correlated with multiple diseases, including Colorectal Cancer, Inflammatory Bowel Disease, and Atherosclerosis[42,44–46]. *Bilophila wadsworthia*, also detected in our dataset, is known to be causative for gastrointestinal discomfort and inflammation[47]. There were also some bacterial associations with environmental variables. We observed other statistically significant, environmental indicators of wastewater microbiome content (Supplementary Data 4), including pH (25 positive associations, 17 negative), specific conductivity (15 positive, 0 negative), and dissolved oxygen content (49 positive associations, 11 negative).

We additionally performed a more targeted search for additional potential bacterial pathobionts of interest. We used the most sensitive taxonomic classifier (Kraken2) for this, as it was most likely to pick up low-abundance organisms; however, it is also prone to false positives, so any results would need to be validated with further effort. Regardless, this approach identified numerous potential pathogens potentially filtered out by Xtree's coverage thresholds or not identified by alignment to MetaPhlAn4's marker gene database (Supplementary Fig. 3), including *Campylobacter jejuni, Enterococcus faecium, Shigella sonnei*, and *Bordetella pertussis*, all of which are pathogens of interest to global health organizations, in large part due to their growing resistance to antibiotics[48–50].

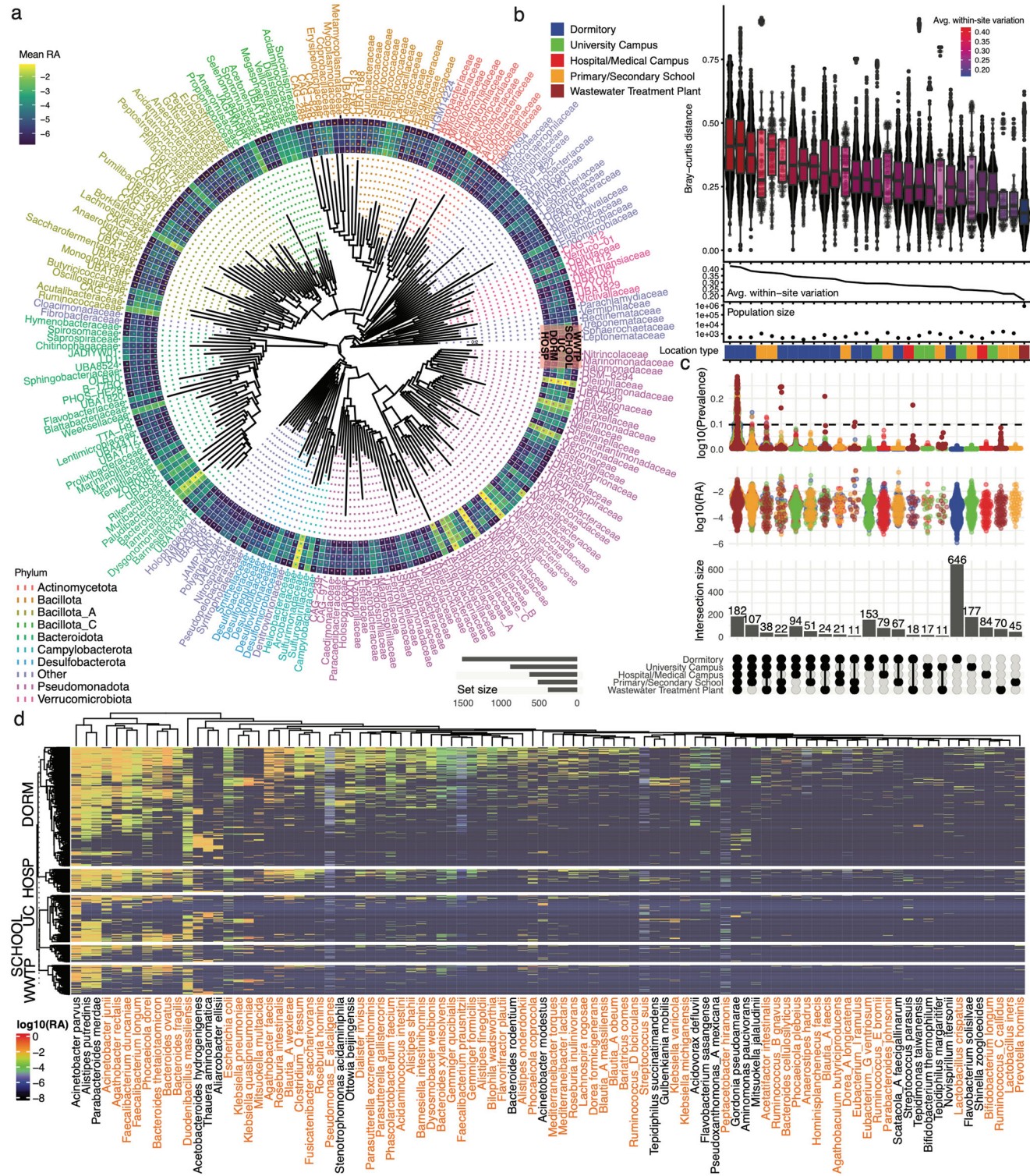

## De novo assembly and short-read alignment provide discrete views of wastewater virome composition

The wastewater virome has been proposed as a potential source for monitoring viral community pathogen load *writ large*[37]. With this in mind, we next interrogated the degree to which the wastewater virome in our dataset could be used for both public health surveillance as well as discovering novel viral genomes. Akin to the bacterial analysis, we implemented multiple methods for viral discovery, using both short-read, kmer-based classification as well as de novo assembly.

Overall, the virome demonstrated even greater within-site diversity than bacterial clades (Supplementary Fig. 4). Despite this, there was a similar trend in relative viral diversity between sites; low-population sites had higher numbers of unique viruses. Only 52 species were detected in every type of sampled location. Viral content had more associations with environmental variables than bacteria (Supplementary Fig. 5, Supplementary Data 4). Detected viruses from short-read alignment derived from families that targeted a diverse array of eukaryotic and prokaryotic hosts and spanned genome compositions (ssRNA+, dsRNA, ssRNA-RT, and dsDNA, Fig. 4a), with the

**Fig. 3 | Wastewater bacterial phylogenetics and diversity. a** The average relative abundance (RA) of all bacterial families represented in the metatranscriptomic data. Outer rings correspond to mean abundance within a family across the different location types extending from the WWTP wastewater treatment plant (outermost ring) to SCHOOL primary/secondary school, UC university campus, DORM dormitory, and HOSP hospital, (innermost ring). The color of the family name corresponds to its phylum. **b** The top panel is the beta diversity (Bray Curtis distance) between all samples within a given site. Each point in this panel represents a comparison between two different samples taken from the same location (e.g., two random samples from a given hospital). The line underneath the top panel corresponds to the average beta diversity at each associated site. The dots in the Population Size sub-panel correspond to the approximate population served by the site from which samples are being compared between. The colored blocks on the bottom are vertically aligned with the population size points and the beta diversity dotplots, and colors correspond to the different sampling location types indicated in the legend. Boxplots represent the median (center line), the 25th (lower bound of the box), and 75th (upper bound of the box) percentiles. The whiskers extend to the smallest and largest values within 1.5 times the interquartile range (IQR) from the lower and upper quartiles, respectively. **c** The intersections between different location types, with the bars indicating the intersection size and the black dots indicating the sites underneath being compared. These bars are vertically aligned with the middle and top panels, which show the relative abundance and prevalence, respectively, of all bacterial species represented by each bar. **d** The log10 RA of bacteria potentially associated with any location type in our Microbial Association Study (MAS). Bacteria occurring in at least three samples and with a BY-adjusted *p*-value of less than 0.1 are plotted. Orange names indicate gut commensals. Source data are provided as a Source Data file.

most prevalent families across sites being *Virgaviridae, Mitoviridae, Tombusviridae, Betaflexiviridae*, and *Picornaviridae*.

After filtering assembled viral contigs for quality and potential viral genes, we identified 913,596 putative genomes of varying quality (903,349 low quality, 5490 medium quality, 4749 high quality, eight complete, Fig. 4b). Median genome length was 350 nt, 3076 nt, 3730 nt, and 5093 nt for low, medium, high, and complete quality genomes, respectively. Clustering these contigs at 90% percent identity resulted in 214,383 non-redundant genomes.

We assigned taxonomy to these contigs and found that they captured a discrete view of the wastewater virome compared to short-read alignment (Fig. 4c); de novo assembly overall outperformed short-read alignment. For example, it captured 46 families and 7 phyla (*Cossaviricota, Duplornaviricota, Negarnaviricota, Nucleocytoviricota, Peploviricota, Preplasmiviricota, Saleviricota*) not detected by alignment. Analogously, both approaches yielded different views of viral genome composition and target host domains: assembly detected expression of ssDNA viruses, alignment did not. Assembly also captured greater phage diversity and more dsDNA viruses, whereas alignment captured more putative eukaryotic-targeting viruses and RNA viruses.

## Wastewater reveals numerous pathogenic viruses and novel viral clades

Given the assembly's ability to provide a taxonomically high-resolution view of wastewater viral communities, we aimed to characterize the phylogeny of assembled RNA viruses (Fig. 4e) using alignments of RNA-dependent RNA polymerase sequences, focusing specifically on the kingdom Orthornavirae. Of the 7058 contigs visualized in Fig. 4e where RdRp could be detected, 6558 (92.9%) belonged to Lenarviricota. This analysis highlighted the challenge of taxonomically annotated (via HMM conservation) genomes; only 165 Lenarviricota genomes had family-level annotations and only 568 (8.0%) of all contigs in this phylogeny even order-level genomad taxonomies, most of them in Picornavirales.

To gain a higher resolution of assembled contigs potentially relevant to human health, we focused specifically on contigs annotated as Pisuviricota, a phylum containing numerous mammalian pathogens (Fig. 4f). To identify potential species-level annotation, we used BLAST to align assembled contigs against all complete genomes in NCBI RefSeq at the 90% identity level, a method utilized in related efforts[51]. To increase confidence in the results displayed, we filtered this tree for contigs above 1000 base pairs. This analysis resulted in the identification of multiple human pathogens, including *Norovirus GII, Sapovirus MC10, Mamastrovirus 1 B1347*, and a *Human astrovirus*. The abundance of these and/or viral strains was present in the short-read alignment data as well. We note that their assembled contigs were annotated as low-quality however their ability to be placed on a reasonable phylogeny, have high homology to reference genomes, and were detected via multiple algorithmic approaches, which indicates that even poor-

quality assembled genomes can feasibly be used for pathogen detection.

We additionally searched the viral assemblies, Kraken2 alignments, and Xtree alignments for viral genomes not represented in Fig. 4 that were of potential public health interest. For example, additional human-relevant viruses (i.e., other *Norovirus* strains, an 8280 base pair putative *Aichivirus* genome) were also annotated by alignment to BLAST but were too short or divergent to be included in the phylogeny (see "Methods" section). Kraken2 was also identified, via short-read alignment.

## Antimicrobial resistance genes in wastewater are associated with hospital antibiotic prescription levels

Having explored targeted RNA sequencing, bacterial metatranscriptomics, and viral metatranscriptomics, we next evaluated the public health benefit of one final microbial data modality: Antimicrobial Resistance Gene (ARGs). Since ARGs are the functional unit of microbial resistance, these data could feasibly provide a high-resolution view into the public health relevance of wastewater sequencing for ARG tracking.

Specifically, we queried the variation in both markers of pathogenicity as well as ARG presence in our dataset. We identified 36,378,395 called Open-Reading Frames (ORFs) in the de novo assemblies. We identified putative ORFs annotated as human health-relevant pathogen-associated proteins[52] as those that varied significantly (adjusted *p*-value < 0.05) across location types as measured by ANOVA (Supplementary Fig. 6). These included various fimbrial proteins, virulence factors, and cell surface proteins involved in infection. We additionally identified other genes associated with microbial-borne disease mechanisms, like CagA, which is the causative gene for *Helicobacter pylori*-caused gastric cancer[53]. Wastewater is hypothesized to be a mode of transmission for *H. pylori* infection[54,55]. We provide, as a genomics resource to the community, our non-redundant protein catalogs clustered at 90% (7,708,163 clustered proteins), 70% (5,171,907 clustered proteins), 50% (3,691,505 clustered proteins), and 30% (3,243,790 clustered proteins) with full annotation data (see "Data availability" section).

We found numerous ARGs throughout our dataset and hypothesized that low-population size sites with high-antibiotic exposure (e.g., hospitals), would have more antimicrobial resistance *writ large* than, say, wastewater treatment plants, where ARGs have been previously identified by other methods predominantly outside of bulk RNA sequencing (e.g., long-read sequencing, culturing)[56–58]. Identification in metatranscriptomics could then identify the activity of these genes as opposed to solely their abundance. Accordingly, we identified an enrichment of these genes overall in hospital samples, compared to all other samples (Fig. 5a). Dormitories had the second-most ARGs, whereas wastewater treatment plants had the fewest. ARGs were consistently the highest in hospitals across all timepoints, and we did not identify any clear seasonality in their monthly variation (Fig. 5b).

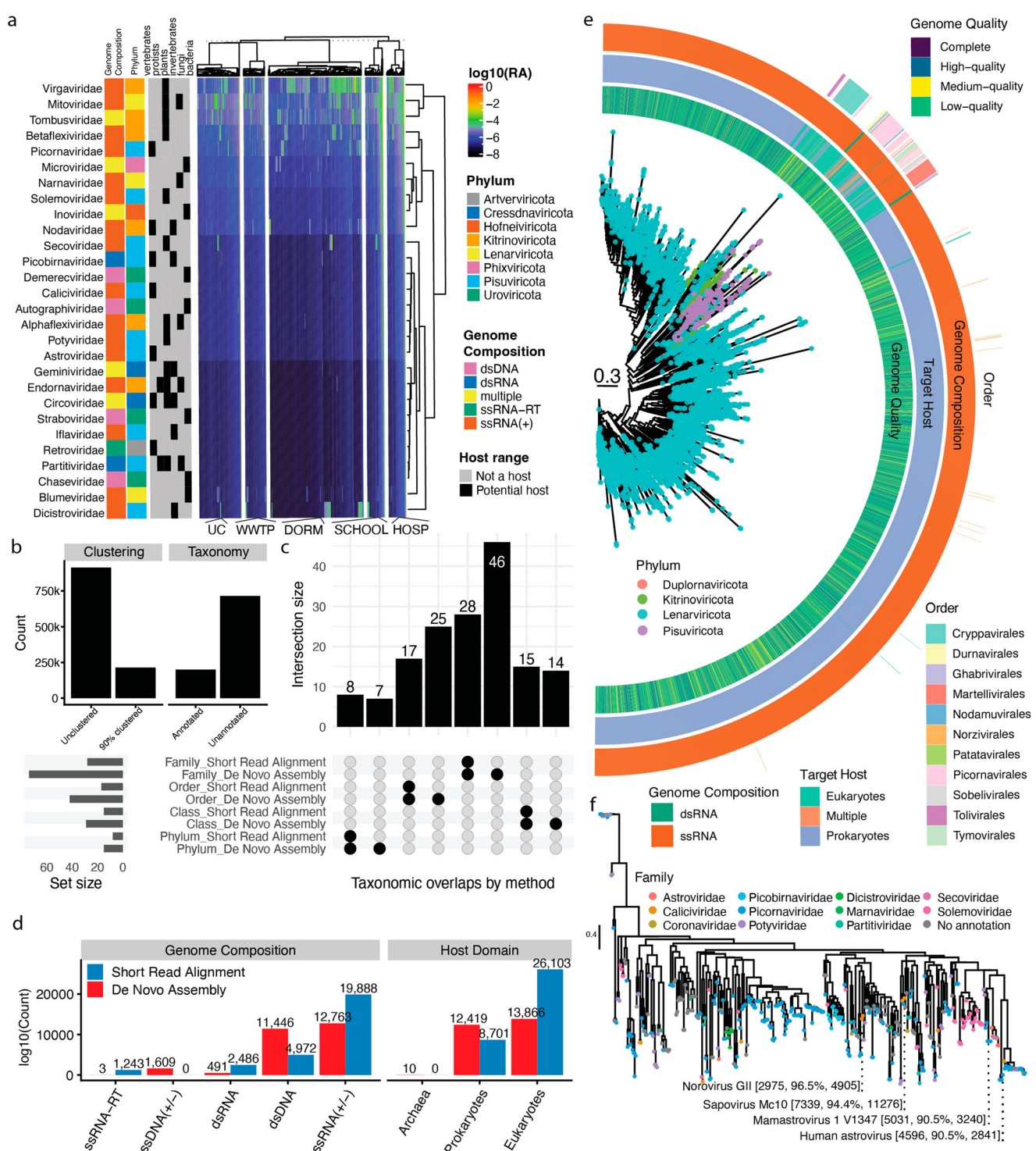

**Fig. 4 | Wastewater viral phylogenetics and diversity through assembly vs. short-read alignment. a** The relative abundance (RA) of top viral families. Heatmap values are log10(RA) of a given viral family as estimated by short-read alignment. Annotation bars on the left-hand side correspond to the International Committee on Taxonomy of Viruses (ICTV) proposed genome composition, geNomad phylum, and ICTV host range for a given geNomad family-level annotation. Alignments were done to a database of 9 public databases comprising 6 million+ dereplicated, taxonomically annotated, and quality-controlled viral genomes (see "Methods" section). Columns and rows are hierarchically clustered. HOSP hospital, DORM dormitory, SCHOOL primary/secondary school, WWTP wastewater treatment plant, UC university campus. **b** Left side: The number of putative viral contigs detected by CheckV compared to the number remaining when clustered at 90% nucleic acid identity. Right side: The number of contigs with and without geNomad taxonomic annotations. **c** The overlap between taxa identified by de novo assembly vs. short-read alignment at different ranks. **d** The different genome compositions and target host information identified by de novo assembly and short-read alignment. **e** A maximum likelihood phylogeny of RNA viruses present in our de novo assembled data. Scale bar is indicated on the plot. **f** A second maximum likelihood phylogeny of RNA viruses present in de novo assembled data annotated as the family *Pisuviricota*. Species-level annotations derive from BLASTing viruses against the complete RefSeq viral genomes at the 90% identity level. The numbers following the species names indicate the genome length, percent identity to the named reference species, and the bitscore of the alignment. Source data are provided as a Source Data file.

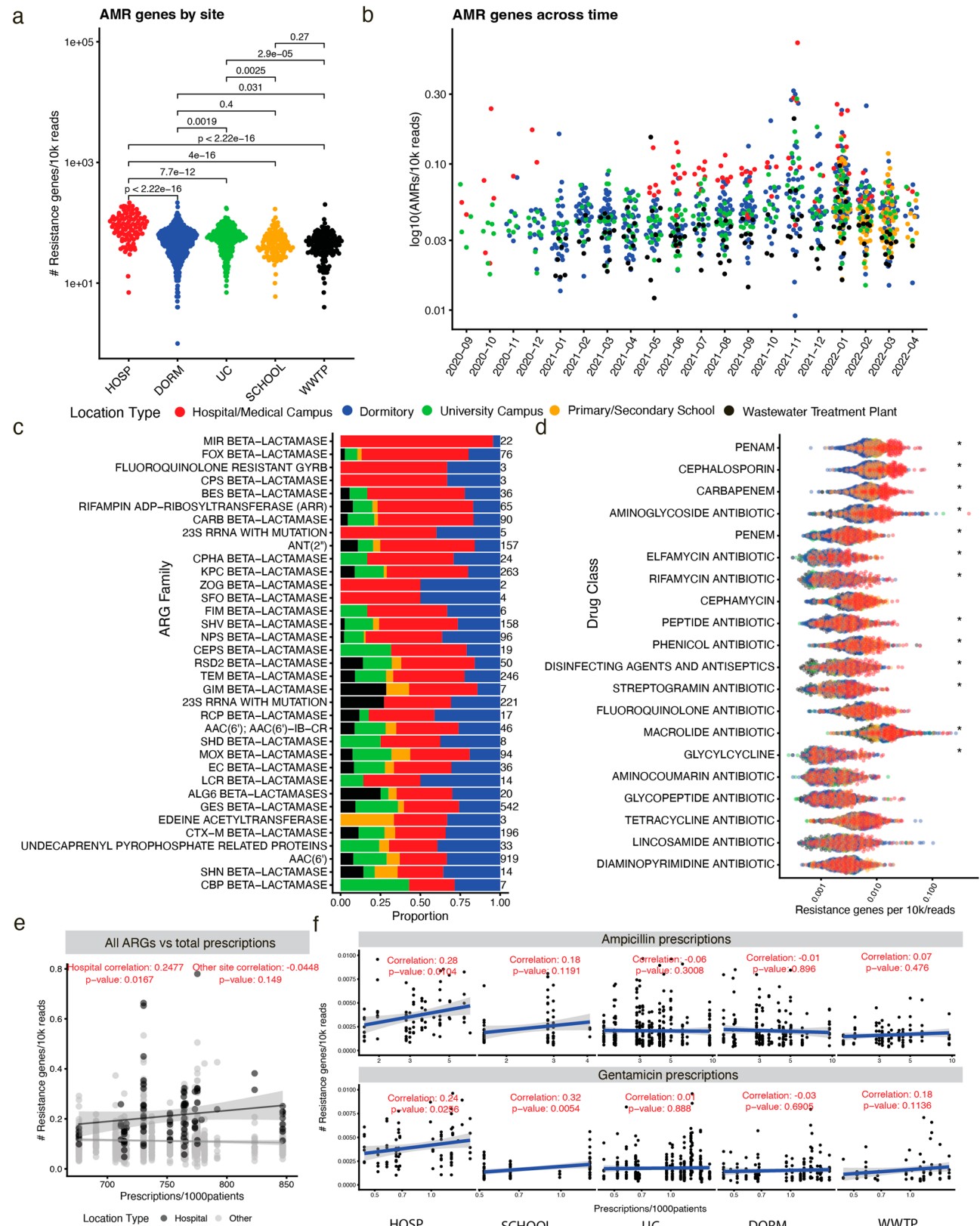

We next evaluated the specific antibiotics and drug classes (Supplementary Fig. 6, Fig. 5c) to which identified ARGs ostensibly conferred resistance. Via t-tests, we identified an enrichment of resistance (adjusted *p*-value < 0.05) to specific types of commonly prescribed antibiotics in hospitals compared to other sites. Tobramycin, Dibekacin, Cefalotin, and Gentamicin resistance dominated in hospitals

(Supplementary Fig. 7). By drug class, penam, cephalosporin, and carbapenem resistance were enriched in hospital wastewater, consistent with the presence of the dominant ARGs. The most prevalent ARGs in hospital wastewater encoded beta-lactamase resistance (Fig. 5d). These genes included GES, KPC, NPS, TEM, CTX-M, CARB, MIR, and FOX.

**Fig. 5 | The antimicrobial resistance landscape of wastewater across time and space. a** The total number of called Antimicrobial Resistance Genes (ARGs) across different sample types per 10 thousand reads, with p-values deriving from *t*-tests on log10 transformed ARG counts normalized by sequencing depth. HOSP hospital, DORM dormitory, SCHOOL primary/secondary school, WWTP wastewater treatment plant, UC university campus. **b** The total ARGs identified across time. **c** The top 25 most prevalent ARGs in hospital wastewater. Color scheme relates to the legend above. **d** The top specific ARGs for different drug classes across all sample types. Asterisks correspond to if a given ARG was enriched (in terms of log10 observations per 10k reads) in hospital wastewater when compared to all other sites according to the adjusted *p*-value on a *t*-test. *P*-values were adjusted by the Benjamini–Yekutieli procedure. For an asterisk to be present, an antibiotic class must have been enriched in all four comparisons (e.g., hospitals vs dormitories, hospitals vs the university campus, hospitals vs primary schools, and hospitals vs the wastewater treatment plant). **e** The Pearson correlation between the sum total of all sampled hospital antibiotic prescriptions and the total ARG counts per 10k for hospitals (black) and (all other sites). This color scheme only applies to this panel. Data are presented as a linear regression line (mean values) surrounded by 95% confidence intervals (area shaded in gray). **f** The same as (**e**), except the only genes considered in the correlation are those annotated as conferring resistance for the two antibiotics listed. The antibiotic data, similarly, is only for prescriptions of those two antibiotics. Data are presented as a linear regression line (mean values) surrounded by 95% confidence intervals (area shaded in gray). Source data are provided as a Source Data file, and all *p*-values reported stem from two-sided tests.

Conversely, certain antibiotics and drug classes (e.g., Kasugamycin, Telithromycin, Capreomycin, Tetracyclines, Glycopeptides) had similar resistance profiles between all wastewater sites. We hypothesize, therefore, that these signs of putative resistance may not derive from exposure to the same concentration of antibiotics as in hospitals, but rather may be indicative of naturally occurring microbial machinery that, while potentially conferring resistance in certain contexts, in this case, are expressed to serve another purpose. Alternatively, specific resistance genes could be deriving from highly similar antibiotics, instead of those named in Supplementary Fig. 7.

Considering that perceived antibiotic resistance may not actually be driven by antibiotic exposure in some cases, we aimed to validate our observation of enriched resistance profiles in hospital wastewater by integrating hospital antibiotic prescription data. We acquired month-by-month prescription data from the hospital sites we sampled in Miami and computed correlations between observed antimicrobial resistance (both overall and on a specific antibiotic basis). To generate a null set of comparisons we computed correlations between prescription data and hospital ARG abundance as well as correlations between prescription data and ARG abundance in all other sites, hypothesizing that non-hospital sites would have limited to no correlation, overall, with prescriptions.

We found statistically significant ($p < 0.05$), moderate positive correlations (Pearson correlation between all antibiotics in hospitals and all ARGs = 0.25) between both (1) overall antibiotic prescriptions and amount of overall hospital wastewater ARGs across all other sites (Fig. 5e) and (2) Ampicillin and Gentamicin prescriptions and their associated specific resistance genes in each specific location type (Fig. 5f). Correlations with Ciproflaxin and Oxacillin were also trending in significance ($p < 0.1$). In other words, the prescription of specific antibiotics correlates to the amount of genes for resistance to those antibiotics in wastewater. Notably, as one would expect, we did not observe these correlations in other sites: this result was unique to hospitals. No correlation was identified between, say, dormitory or wastewater ARG load and hospital prescriptions. We claim this observation strengthens the hypothesis that wastewater ARG levels can predict the spread of antibiotic resistance in a given community based on antibiotic use data (https://www.ncbi.nlm.nih.gov/pmc/articles/PMC6396544/), potentially meaning wastewater could be used to directly advise physician prescriptions and guide antibiotic stewardship.

## Discussion

In this longitudinal wastewater microbial census, we integrated targeted qPCR plus targeted and untargeted RNA sequencing to detail the bacterial, viral, and functional landscape of samples from multiple areas within Miami-Dade County across two years. By combining these different approaches, we capture the breadth and depth of microbial diversity: targeted sequencing, for example, can identify viruses beyond the limit of detection of bulk sequencing (down to 100 genomic copies per liter)[7,31], which itself is superior at measuring overall community composition. Further, we leverage and compare multiple, redundant bioinformatic approaches, demonstrating how different algorithms (assembly vs. short-read alignment, for example), highlight distinct site-specific and shared microbial features. With the increased integration of wastewater sequencing and geospatial epidemiologic data, we can identify increasingly precise signatures for these and other features, potentially replacing onerous and costly public health surveillance methods in the long run.

Overall, we showed that wastewater can be potentially used for (1) viral pathogen detection, (2) observing gut-associated microorganisms positively and negatively associated with human health, and (3) tracking geospatially varying and prescription-correlated, drug class-specific, community antimicrobial resistance. In addition to various pathogens of interest, we detected microbes known to be found in the human gut that have reported associations with human health and human habits, including *F. prausnitzii, R. hominis*, and *B. wadsworthia*. Finally, resistance genes for fluoroquinolones, cephalosporins, penams, aminoglycosides, macrolides, and glycylcyclines antibiotics were detected at high concentrations in hospital wastewater samples. Many of the specific genes we identified are plasmid-encoded enzymes and are responsible for the rise of resistant infections worldwide[59].

As a result, based on the current needs within public health surveillance and the results of this (and other) studies, we propose two immediate courses of action for extending these efforts and building tools that leverage wastewater microbiome ecology to track public health: (1) using wastewater to mitigate community antibiotic resistance spread by, for example, advising physician prescriptions based ARG wastewater abundance (comparing ARGs detected in wastewater sequencing to existing approaches, like hospital-specific antibiograms) and (2) monitoring pathogens and determining the feasibility of constructing gut health indicators of community diet and overall disease prevalence. This could include, for example, exploring the potential to discover biomarkers of cancer. Regarding the former, new antimicrobial resistance markers could feasibly be discovered by correlation prescription levels with de novo assembled gene abundance in wastewater. Regarding the latter: for validation and biomarker development, existing, zip code-stratified public health surveillance datasets, like the National Health Administration Nutrition Examination Survey (NHANES)[60], should be integrated with ongoing wastewater sequencing in the same zip codes to identify wastewater-based correlates of the variables (e.g., lab tests, dietary survey data, disease burden) recorded in NHANES. Together, these two initiatives could provide a tool to mitigate antibiotic resistance in real-time while also tracking public health and nutrition at high resolution (e.g., monitoring the spread of food deserts).

To take advantage of wastewater's potential as a single point source for monitoring multiple population-level characteristics, uniting both targeted (e.g., specific pathogens or resistance markers) and untargeted analyses (metagenomics or metatranscriptomics) will be critical (https://www.sciencedirect.com/science/article/pii/S0048969721042509). Targeted ARTIC sequencing, for example,

can identify extremely low-abundance (parts per billion) viruses that require many rounds of amplification via PCR to be able to sequence. It could feasibly also be used to target specific disease-causing microbial genes, like CagA (in the case of gastrointestinal cancer). However, it is limited in its ability to only capture a single organism per set of primers. Bulk sequencing, on the other hand, can theoretically capture information from all domains of life living in wastewater. Given that wastewater is an aggregate of environmental and human biology from a single geography, it stands to reason that microbial signals in wastewater could be used as a one-stop-shop for public health surveillance. From heavy metal levels (e.g., lead, copper, fertilizer runoff) to gut health to cancer clusters, theoretically, integrated microbial surveillance could capture all modalities in one sample type for low cost, with limited privacy concerns.

An added benefit of the wastewater surveillance documented here, specifically, is the continuous monitoring of geographically distinct sites serving different community sizes. We observed variation in microbial composition and consistency across represented population size, for example, observing low variability in wastewater treatment plant microbiomes compared to dormitories. It stands to reason that temporal variation in major environmental pollutants might be best monitored at treatment plants, whereas isolated outbreaks (e.g., gastroenteritis) might be better tracked at locations housing smaller populations.

Of course, our approach is not without drawbacks. Some SARS-CoV-2 genomes were likely filtered out due to our coverage threshold, some wastewater samples had low depth of sequencing, and a certain amount of data could be lost prior to collection or during processing; this can dramatically confound associations. Similarly, our processing methods (electronegative filtration, etc), could skew microbial community composition. Further, microbes from other hosts (e.g., farm animals) that appear human-associated could add noise to public health signals, so monitoring and controlling sampling locations carefully will be critical[61]. Analogously, some genes – like ARGs – that appear human health-relevant may in fact just be naturally occurring and not utile biomarkers. Additionally, bioinformatic annotation of viruses is extremely challenging; as we observed, species-level resolution is difficult to attain and can yield false positives. Even BLASTing at a high percent identity (>90% in our case) could yield misclassifications at the species level. Finally, we focus here on viruses and bacteria; fungi are also important for human health and detectable in wastewater[62,63]. Feasibly, this dataset we generated could be used, in future work, to measure fungal pathogen abundance as well.

In total, by uniting targeted pathogen tracking with broad monitoring of epidemiologically relevant microbial and environmental signals, there is a potential to use wastewater as a primary source for monitoring multiple dimensions of public health. The development of consistent protocols for sampling, sequencing, and analyzing wastewater data is critical for it to achieve clinical impact (https://pubs.acs.org/doi/full/10.1021/acsestwater.2c00045). Further, once these protocols are established, robust biomarkers for human population health must be discovered via large-scale association studies, leveraging those that have been successful in other disciplines[42,64–67]. This study represents an additional step in this direction. Overall, by characterizing the ecology and its variation across time and space, we aimed to lay the groundwork for building a dataset of taxa of interest to be monitored with bulk RNA sequencing alongside targeted approaches for pathogens of import. While shotgun metagenomics and metranscriptomics methods hold rich information about a wide range of pathogens, they can also be used to monitor host dynamics, drug interactions, (https://pubmed.ncbi.nlm.nih.gov/33712587/), and eukaryotic species as well (https://pubmed.ncbi.nlm.nih.gov/26836631/). Future studies can link to changes in other organisms' movements and genotypes (e.g. zoonotic reservoirs), and also link to national-level wastewater profiling,

such as the CDC's National Wastewater Surveillance System (NWSS). Eventually, linking regional, national, and international datasets would empower a global, semi-automated surveillance system, which could serve as a first line of defense against future pandemics, contaminant buildup, ARG tracking, and establish essential tools to mitigate future public health crises.

## Methods

### Study design and IRB approval

The goal of the study was to evaluate associations between clinical data and wastewater SARS-CoV-2 RNA levels in matched populations as well as describe the microbial composition of wastewater via bulk RNA sequencing. This project was approved by the University of Miami IRB (#20210164). Starting the Fall 2020 semester, to optimize on-campus safety against the spread of COVID-19, the University of Miami embarked on a clinical tracking program based on testing, tracking, and tracing (3-T). For students who lived or commuted to campus, the University required weekly or bi-weekly testing. Employees were tested on a random basis.

All patient data used in this study was published in a separate manuscript, Carattini et al.[68]. Documented clinical data were available on a building basis at the University of Miami dormitories, for the University campus for both student and employee populations who resided or commuted to campus, and at the University of Miami hospital. In addition, a subset of positive clinical samples (both students and employees) were amplified for SARS-CoV-2 variants and sequenced at a University of Miami laboratory (Oncogenomics Shared Resource, OGSR). The OGSR ran both the clinical and wastewater samples reported in this study providing for consistency in the detection between both datasets. The sharing of clinical data for internal research purposes was approved by the University of Miami IRB (IRB ID: 20210164, MOD00047150). In addition, at the county level, the Florida Department of Health documented positive COVID-19 detections by zip code and this data was provided to the research team through delegation of the above IRB. Given the availability of clinically-based data, the wastewater sample collection program was designed to provide a population match between clinical data and populations contributing wastewater to a specific wastewater sampling location. A waiver of consent was obtained for the retrospective review of charts and test results, and for the review of aggregated and deidentified data inclusive of SARS-CoV-2 sequencing results from environmental and clinical samples.

### Sample collection

Samples were collected weekly from September 30, 2020, through September 21, 2022. Sampling sites included sewer holes from residential dormitories (14 sites), sites representing major portions of campus (4 sites, 3 corresponding to the main residential campus and 1 corresponding to the university marine campus with no residences), from laboratory/administrative buildings at the medical campus (2 sites), from the University Hospital (2 sites), grade school sites (8 locations representing 9 grade schools, 3 high schools, 2 middle schools, and 4 elementary schools with one elementary school and one middle school discharging to the same sewer hole), and from a regional wastewater treatment plant (Central District) serving a population of 830,000 from Miami-Dade County. In addition, at three sites both grab and composite samples were collected accounting for the balance of the 34 sampling sites. Between September 2021 and January 2022, samples were collected two times per week at the dormitory sites to provide additional data for on-campus mitigation measures.

Composite samples were collected at the building and cluster scale using either an ISCO 6712 autosampler (time-paced) or at the community-scale wastewater treatment plant using a HACH auto-sampler (flow-paced). All other samples were collected using a grab

sampling technique, due to the limited availability of multiple auto-samplers. Sample collection volumes were 2 liters with the first liter being used for water quality analyses in the field (pH, water temperature, dissolved oxygen, specific conductivity, and turbidity). Additional details about sample collection and basic water quality data have been published earlier[7,29,30,33].

**Primary concentration and downstream processing of wastewater.** Upon collection, wastewater was transported to the Biospecimen Shared Resource Laboratory at the University of Miami on ice and immediately succumbed to pretreatment for a primary concentration of suspended solids. Wastewater samples had [10^6 genomic copies (gc)/L] human coronavirus-OC43 added as a recovery tool for quantitative PCR assessment, occurring prior to targeted sequencing. Additionally, 51% w/v magnesium chloride was added as well as drops of 10% Hydrochloric Acid (pH reduction down to 3.5–4.5) to assist with altering the chemical charge of ambient viral particles found in wastewater to positive. The primary concentration method employed was Electronegative Filtration (ENF)[31] using HAWP mixed cellulose ester membranes (Millipore Sigma #HAWP04700, 47 mm diameter, 0.45 μm pore size) which were negatively charged, thus requiring the necessity for the pretreatment to aid in binding affinity between the viral particles and membranes. Via the use of vacuum filtration, electronegative HAWP membranes captured wastewater-suspended solids, and variable sample volumes (20–150 mL, depending upon turbidity and water quality parameters) were filtered until clogging/filter saturation to create concentrates. HAWP filter membranes containing a saturated layer of suspended wastewater solids were folded in on themselves four times, and placed immediately within 1.5 mL 1× DNA/RNA shield within 5 mL microcentrifuge tubes. Three filter concentrate replicates were generated per wastewater sample collected over the course of the study period; each replicate was sent to a separate laboratory, the Center for AIDS Research (CFAR) Laboratory (on ice), the OGSR Laboratory (on ice) both at the University of Miami and at Weill Cornell Medicine (WCM) (on dry ice).

Three workflows were implemented to extract SARS-CoV-2 from wastewater which occurred in separate laboratories to standardize and achieve overlapping analyses following extraction. These processes included quantitative PCR (at CFAR), ARTIC analysis (at the OGSR), and bulk RNA sequencing (at WCM). At the CFAR laboratory the Zymo Research QuickRNA-Viral Kit, with a modified protocol using 250 μL wastewater concentrate - discussed in detail within Babler et al.[29] to reduce inhibition - was utilized as input for Volcano 2nd Generation-qPCR (V2G-qPCR). This in-house created assay is equivalent to that of RT-qPCR but bypasses the need for the initial reverse transcription step as the DNA polymerase utilized within the reaction can read both RNA and DNA templates, thus reducing the total reaction time and allowing for additional analyses of other targets within the same timeframe (i.e., normalization parameters, control parameters)[7]. To assess for inhibition within RNA samples, 30 μL of HIV RNA was added to each 10 μL eluate of wastewater RNA, and assessed alongside a weekly generated water control (10 μL Nuclease-Free water + 30 μL HIV RNA) by V2G-qPCR; following amplification, the Cq values were compared and if the wastewater sample Cq values measured within ±2 cycles of the water control, samples were considered uninhibited.

The choice for determining which samples were ultimately brought through ARTIC analyses, via collaborative communication between CFAR and OGSR, was based on the gc/L measurements determined by V2G-qPCR taking into consideration the overall recovery and inhibition as determined by the controls. This was because routine extraction and multiple V2G-qPCR assessments occurred within the CFAR laboratory weekly, wherein the filter replicate sent to the OGSR following concentration was kept at −80 °C until extracted in batches for specific analyses.

## Targeted sequencing of SARS-CoV-2

Libraries were synthesized using the NEBNext® ARTIC SARS-CoV-2 FS Library Prep Kit (E7658L) following the Express Protocol and employing the V3 primer set throughout. Eight microliters of total nucleic acid (TNA) extracted with Nanotrap Microbiome Particles (Ceres Nanoscience, 44202) enrichment and MagMAX™ Viral/Pathogen Ultra Nucleic Acid Isolation Kit (ThermoFisher ScientificSciencific, A42356) were used as input without quantification nor assessment of RNA integrity. Finished libraries were pooled volumetrically with final pools of 96 libraries cleaned using 0.9× AMPure bead cleanup (Beckman A63882). Pools were sequenced on an Illumina NextSeq 500 using Mid-Output 150 cycle flow cells (130 M clusters, 20024904) run as 76|8|8|76 chemistry or NovaSeq 6000 using various flow (SP S1, S2 or S4; 20028400, 20028319, 20028317 20028314, or 20028312, respectively) as 151|8|8|151 chemistry, with up to 3% PhIX (FC-110-3001).

## Detection of SARS-CoV-2 variants in ARTIC data

Broadly speaking, the pipeline for variant detection involved (1) identification of reads aligning to the SARS-CoV-2 genome with Kraken2, (2) aligning to the Wuhan-Hu-1 reference and filtering samples based on breadth and depth of alignment coverage, (3) trimming primers, (4) calling variants using an ensemble approach, (5) annotating mutations, and (8) estimating VOC lineage abundances. All software was run with the default settings unless otherwise specified.

We used Kraken2[69] (V2.1.2) running the default settings to taxonomically classify short-read sequencing data for the ARTIC samples. We used a custom Kraken2 database that included the SARS-CoV-2 Wuhan-Hu-1 reference (GCF_009858895) as well as an assortment of viral, archaeal, bacterial, fungal, protozoan, and mammalian genomes (available upon request). The reads assigned exclusively as SARS-CoV-2 were then filtered into individual files using seqtk (V1.3-r106). Alignment to the Wuhan-Hu-1 reference was done with bwa (V0.7.17-r1188), and the alignment was then sorted and indexed with sambamba. We then trimmed primers with ivar (V1.13) and generated coverage statistics afterward with the bedtools genomecov (V2.30.0) command and mosdepth (V0.3.3). We filtered out samples that did not have at minimum 10× mean coverage per amplicon across at least 73 of the 98 amplicons (roughly 75%) that were targeted by the ARTIC protocol.

For variant calling, we used an ensemble approach, combining the output of lofreq (V2.1.5), and iVar, only the union of calls found in both approaches, and computing the mean variant allele frequency between the two. We used VEP (V104.3) to annotate mutations and estimate VOC lineage abundance with Freyja (V1.4.2), using the latest available VOC database (as of April 2023), provided by the software.

## Bulk RNA sequencing

After concentration via ENF, the filter concentrates were preserved in 1.5 mL of 1× DNA/RNA Shield (Zymo Research, R1100), transported on dry ice, and remained stored at −80 °C until further processing at Weill Cornell Medical College. Nucleic acid extraction took place using the DreamPrep NAP workstation (TECAN, NAP-DREAMPREP), a liquid-handling automation device, and the Quick-DNA/RNA Viral Magbead Kit (Zymo Research, R2140) using 400 μL of wastewater concentrate, following manufacturer's recommendations.

Samples were sequenced in two batches by two separate providers. HudsonAlpha Discovery (Huntsville, AL) performed the first sequencing library preparation on the generated nucleic acids. The nucleic acid samples underwent purification using the RNA Clean and Concentrator Magbead Kit with DNase digestion (Zymo Research, R1082), following the manufacturer's recommended protocol on the Biomek i5 automated workstation (Beckman Coulter, 52018). This process resulted in an 18 μL elution of RNA in nuclease-free water. The LabTouch GX Touch (Perkin Elmer, CLS137031/C) nucleic acid analysis

system quantified the RNA, with samples prepared according to the manufacturer's instructions.

After normalizing RNA concentrations, sequencing libraries were generated using the NEBNext rRNA Depletion Kit (Human/Mouse/Rat) and NEBNext Ultra II DNA Library Prep Kit (New England Biolabs, E6310X, E7645L) for Illumina. The KAPA Library Quantification Kit (Roche, 07960140001) was employed to quantify the resulting libraries, all samples were brought to 20 cycles of PCR at the final step. All libraries were next evaluated and pooled at equimolar concentrations. Sequencing was carried out on the NovaSeq 6000 platform (Illumina, 20012850) with an S4 flow cell, using a read length of 2 × 151 bp in paired-end configuration.

The second batch of samples were sequenced by Element Biosciences. The nucleic acid samples underwent purification identical to that of HudsonAlpha Discovery, using the RNA Clean and Concentrator Magbead Kit with DNase digestion (Zymo Research, R1082), which cuts both double-stranded and single-stranded DNA. The resulting elution of RNA was quantified using a Qubit 4.0 Fluorometer (ThermoFisher, Q33238), using 10 ng of total input for the library preparation. The sequencing libraries were generated using the NEBNext rRNA Depletion Kit (Human/Mouse/Rat) and NEBNext Ultra II DNA Library Prep Kit (New England Biolabs, E6310X, E7645L) for Illumina, following the vendor protocol with two modifications. First, Element Biosciences adapters and unique dual indexes (Element Biosciences, 830-00005) were used instead of NEB-recommended Illumina adapters and indexes. Second, because of this change, the USER treatment step was omitted. Completed libraries were quantified by Qubit or equivalent and run on a Bioanalyzer or equivalent for size determination. Libraries were pooled and then circularized and quantified using the Element Elevate Library Circularization Kit (Element Biosciences, 830-00001). Sequencing was performed using the Element AVITI sequencing instrument (Element Biosciences, AVITI), using 2 × 75 bp high output kits (Element Biosciences, 860-00004) run in HD Expert Mode.

## Metatranscriptomic quality control and short-read taxonomic profiling

We submitted all short-read metatranscriptomic sequencing data first to a quality control pipeline, predominantly using the bbtools suite (V38.92)[70]. Clumpify (parameters: optical=f, dupesubs=2,dedupe=t) was used to group overlapping reads, and bbduk (parameters: qout=33 trd=t hdist=1 k=27 ktrim="r" mink=8 overwrite=true trimq=10 qtrim='rl' threads=10 minlength=51 maxns=−1 minbasefrequency=0.05 ecco=f) was used to deduplicate reads and remove adapters, accounting for duplicates generated by the PCR-amplication process. Potential human contaminating reads were removed by bowtie2 (V2.4.4, parameters: --very-sensitive-local) alignment to the HG38 human genome assembly[71]. Any samples remaining with uneven numbers of reads were repaired with bbtools' repair function. Finally, tadpole (parameters: mode=correct, ecc=t, ecco=t) was used to correct sequencing errors.

Quality-controlled reads were run through different taxonomic classification approaches: (1) Kraken2 (V2.1.2) against all of the complete genomes for microorganisms on RefSeq (2) and MetaPhlan4 (V4.0.4) against its default database, (3) Xtree (V0.92i) against the Genome Taxonomy Database (GTDB) r207, and (4) Xtree against the dataset of complete a 6 million dereplicated, high-quality viral genomes (the Pan Viral Compendium) from nine public databases, the metadata of which is available at Figshare (https://figshare.com/articles/dataset/PVC_Release_V0_1/24566995/1). All of the genomes in these listed databases were dereplicated at the 90% identity level and quality checked and annotated with CheckV and geNomad (identically as described below in the de novo assembly section). Alignments were done to the database of dereplicated complete, high, and medium-quality genomes.

We ran Kraken2 with both –confidence 0.2 and –confidence 0.0 flags; the latter parameter was only used for estimating total, domain-level read alignment. The former was used for species-level annotations (e.g., as presented in Supplementary Fig. 3). Abundance matrices for both confidence flags are available online (see "Data availability" section).

Xtree is a kmer-based aligner that generates coverage statistics (global, which corresponds to total genome coverage, and unique, which corresponds to reads mapping to a specific genome in the database). We filtered for bacterial genomes with 1% global and 0.5% unique coverage. We filtered for viral genomes with 20% global and 10% unique coverage. Relative abundances for Xtree were computed by dividing the total aligned reads to a given genome over the total number of reads aligning to all genomes. MetaPhlAn4 computed relative abundance automatically. We used Bracken to compute relative abundances for the kraken2 output. All the figures in the text were generated with the Xtree relative abundances unless otherwise specified.

The bacterial phylogeny generated in Fig. 2a was generated by subsetting the GTDB taxonomic tree (provided at https://data.gtdb.ecogenomic.org/releases/release214/214.1/) to the species identified in the metatranscriptomic data. Family-level representatives were selected at random. Their abundance, reported in the circular heatmap, was computed by averaging the relative abundance of all species in that given family.

## Removal of potential kitome or sequencing contaminants

To account for potential contaminants from sample processing, we additionally sequenced and bioinformatically leveraged 15 negative controls. These were uninoculated DNA preservative buffered (DNA/RNA Shield) taken during extraction that was sequenced alongside the other samples. We used the decontam R package (isContaminant function, parameters: method="prevalence", threshold=0.5) on all relative abundance matrices generated by the above alignment strategies[72]. We used the parameters and approach derived from the following example: https://benjjneb.github.io/decontam/vignettes/decontam_intro.html. Taxa removed during decontamination from each matrix are reported in Supplementary Data 2. Both decontaminated and raw abundance matrices are available (see "Data availability" section).

## Microbial Association Study

The Microbial Association Study (MAS) reported in Fig. 3 and Supplementary Figs. 2 and 4 was executed via a linear mixed modeling approach. Relative abundances were log-transformed with the smallest non-zero abundance value being added to the entire matrix beforehand. The following mixed effects model was fit for each bacterial/viral species that was found in over 10 samples:

$$\log 10(microbial\ feature) \sim feature + (1|locationID)$$

The independent variable is the log10 transformed abundance of any given bacteria and the dependent "feature" variables are the features described in Supplementary Figs. 2 and 4 and Supplementary Data 4. Note that log10 transformation worked slightly differently for each aligner based on the raw data they output. MetaPhlan4 and Bracken give relative abundances (ranging from 0 to 100 for the former, and from 0 to 1 for the latter); for both, we added a pseudocount of 0.000001 to all zeros. Xtree gives raw counts. For it, we replaced all zero values prior to computing relative abundance with 0.5 and then log10 transformed. P-values were adjusted via the Benjamini–Yekutieli procedure[73], and an adjusted cutoff of 0.05 was used to gauge significance.

## De novo assembly, viral contig identification and annotation, gene catalog construction, and Resistance Gene/Open-Reading-Frame identification

Quality-controlled reads were de novo assembled using MetaViralSPAdes[74] V3.15.5. The soft filtered transcripts output by MetaViralSPAdes were fed into CheckV V0.7.0 to predict putative viral contigs[75]. For all high, complete, medium, and low-quality contigs (i.e., putative viral contigs) Open-Reading-Frames (ORFs) were identified with Prodigal-gv, a version of Prodigal[76] modified for viral ORF identification (https://github.com/apcamargo/prodigal-gv?tab=readme-ov-file)[77]. Called ORFs were annotated against the pFam database (most recent version as of Feb 1, 2024) using hmmsearch[78] (parameters: --notextw --noali --nobias --nseq_buffer 100,000 --nhmm_buffer 1000). The best pFam annotation (based on minimum e-value) for each gene was assigned. ORFs were clustered into a non-redundant gene catalog at 90%, 70%, 50%, and 30% identity using mmseqs2 (V15.6f452, parameters: -c 0.9 --cov-mode 1) Antimicrobial Resistance Genes (ARGs) and virulence factors were identified with RGI (V6.0.3, parameters: –low_quality, –num_thresholds 4, –include_nudge) using the CARD database (most recent versions as of Feb 1st, 2024)[79]. We report in Fig. 5 the numbers of ARGs per 10,000 reads sequenced to account for variation in sequencing depth between samples.

Putative viral contigs were annotated first with geNomad[77]. Specifically, for RNA viruses used in the phylogenies in Fig. 4, we additionally implemented a BLAST[80] approach for species-level annotation, modeled directly on published methods[51], with code available at https://github.com/snayfach/MGV/blob/master/ani_cluster/README.md. Briefly, we BLASTed putative viral contigs against the RefSeq database at the 90% identity level (using the lines and scripts provided in the Github Repository). We report 4 matches, as well as contig lengths and bitscores, in Fig. 4.

## Viral phylogeny construction

We additionally constructed RdRp viral trees for viruses annotated in the *Orthornavirae* kingdom by geNomad. We used BLAST to cluster the 61,557 contigs with this annotation at the 90% identity level using the approach described above[51]. This resulted in 10,446 genomes. For each of these, we first identified RdRp sequences using a recently published database and HMMER3[78,81]. We took all annotations with an e-value under 0.01. Viral RdRp ORFs which contain hits from several regions were concatenated into one continuous genetic sequence per sample and merged into a single FASTA file. Filtering based on e-values resulted in a total of 8131 contigs containing putative RdRp sequences.

MAFFT[82] was used to align viral sequences. Due to the nature of viral sequence alignment, gappy regions were filtered from our alignment in this case we trimmed columns that exceeded 67% using TrimAl[83]. RAxML[84] was used to infer our phylogenetic tree using the LG evolutionary model[85] with 10 starting parsimony trees and 100 bootstrap replicates in order to obtain support values for our topology. When plotting, to prune the tree and remove highly divergent viruses, we removed branches in the top 5% of lengths. We additionally identified the most prevalent phylum annotation in each clade (10 nodes back from a given tip) and removed genomes that were outside of that annotation for said clade. This, combined with the length trimming, reduced the total number of genomes included in the tree in Fig. 4e to 7058.

We took the same approach for generating the tree in Fig. 4f. We took the subset of 90% BLAST dereplicated viruses annotated as *Pisuviricota* and used RAxML with the same parameters to place them on a phylogeny. The same pruning approach was taken for tree visualization.

## Analysis of pathogenic genes in wastewater

We filtered ORFs (unclustered) for those with pFam annotations with e-values below 0.01. For each annotation, we fit an ANOVA to identify those varying in prevalence (i.e., count of a given annotation in a given sample) between location types (primary/secondary schools, the wastewater treatment plant, hospital/medical campus sites, the university campus basin, and dormitories). We only considered annotations that had prevalences above 25 across all samples (9177 pFam annotations). We adjusted p-values with the Benjamini–Yekutieli procedure and filtered for those with adjusted p-values under 0.05 ($N = 4380$). We then intersected these significant pFam annotations with the pFam annotations used in the PathFam database[86], filtering for those annotations that were significant in our analysis and, in PathFam, had a hypergeometric test adjusted p-value of less than 0.05 and also occurred in both the VFDB and Victors databases ($N = 102$). The resultant protein prevalences are visualized in Supplementary Fig. 6.

## Association between Antimicrobial Resistance Genes and hospital prescription data

We sourced monthly hospital antibiotic prescription data (per 1000 patients) abstracted from the health system's electronic health record, Epic. We computed the total number of identified ARGs by month (overall and by the antibiotic to which a given gene conferred resistance). We took the intersection of antibiotic prescribed and genes annotated as having resistance to those antibiotics and computed a Pearson correlation between the log10 counts of prescriptions and the number of genes per 10,000 sequencing reads. We removed samples with fewer than 1 million reads for this analysis, as we found they artificially inflated the number of ARGs identified. We report raw p-values in the figures and text. We computed and reported correlations between both hospital ARG totals as well as the other sites sampled.

## Additional software

All analysis was done in RV4.1.3[87]. Additional packages used include: ggtree V3.7.2[88], ggplot2 V3.4.2[89], the tidyverse V2.0.0[90], ComplexHeatmap V2.10.0[91], UpSetR V1.4.0[92], ComplexUpSet V1.3.3[93], phytools V1.5-1[94], ape V5.7-1[95], ggbeeswarm V0.7.1, tidytext V0.4.1[96], circlize V0.4.15[97], vegan V2.6-4[98], broom V1.0.4[99], reshape2 V1.4.4[100], and ggpubr V0.6.0[101].

## Reporting summary

Further information on research design is available in the Nature Portfolio Reporting Summary linked to this article.

# Data availability

In compliance with the NIH RADx-rad Data Coordination Center (DCC) requirements, the raw sequencing data was submitted to the Sequence Read Archive (SRA). The wastewater samples were annotated with the rich metadata and the sequencing information (fastq files) was included in the submission. The submitted data can be found in the SRA under the accession PRJNA946141. Source data are provided with this paper. Patient data used in Fig. 1 are described in the following manuscript, Carattini et al.[68]. Furthermore, the metadata associated with the wastewater samples' sequencing data was extracted from the Illumina operational files, validated, organized, and submitted to the NIH data hub via DCC [https://radx-hub.nih.gov/home], where the SF-RAD data is associated with the dbGaP study accession phs002525.v1.p1. The metadata standards specifications used to describe the data were developed in collaboration with the SF-RAD members and the DCC and formally defined and registered at FAIRsharing.org. Additional processed files (e.g., taxonomic abundance matrices) are available at https://figshare.com/projects/Geospatially-resolved_public-health_surveillance_via_wastewater_sequencing/198412. Source data are provided with this paper.

## Code availability

All software used were published tools (which we report the parameters and implementations for in the other "Methods" sections). Scripts used for plotting and slurm-based job management are available at https://github.com/b-tierney/radx-wastewater-scripts/tree/main[102].

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

## Acknowledgements

This study was financially supported by the National Institute on Drug Abuse of the National Institutes of Health (NIH) under Award Number U01DA053941, R01AI151059, WorldQuant, and the GI Research Foundation (GIRF). The content is solely the responsibility of the authors and does not necessarily represent the official views of the NIH.

## Author contributions

C.E.M., H.M.S., and G.S.G. conceived the study. They coordinated wastewater sampling alongside all authors based at the University of Miami. D.A. and Y.C. provided results from clinically derived samples. S.L.W. and B.C. additionally led the sequencing of the targeted amplicon data for wastewater and clinical samples. Authors from Weill Cornell (K.A.R., D.B., N.D., B.G.Y., C.M., A.G.L., J.W.H., J.P., and K.C.K.) led the sample extraction, sequencing, and analysis of the bulk metatranscriptomic data. Wastewater sample collection was designed and facilitated by K.M.B., A.A., B.R., J.L., S.C., W.E.L., J.J.T., N.K., and H.M.S. Targeted PCR was designed and facilitated by M.E.S., K.M.B., and A.A. Human subjects compliance was facilitated by N.S.S., C.C.B., and E.K. Facilities were available through M.S., M.M.B., and G.S.G. Data standards and data archiving were facilitated by D.V., S.C.S., and X.Y. Clinical-based information from the hospital was available through B.S. B.T.T. led the analysis and wrote the manuscript with C.E.M. and H.M.S. JF designed the amplicon sequencing bioinformatic pipeline. All authors (including G.A.G. and G.M.C.) contributed to writing and editing the manuscript as well as providing guidance on analytic choices and validations.

## Competing interests

B.T.T. is compensated for consulting with Seed Health on microbiome study design. C.E.M. is a co-founder of Onegevity and Biotia. No entity listed here was involved in funding or advising the contents of this study. G.M.C. lists competing interests at arep.med.harvard.edu/tech.html. The remaining authors declare no competing interests.

## Additional information

[1]Department of Physiology and Biophysics, Weill Cornell Medicine, New York, NY, USA. [2]The HRH Prince Alwaleed Bin Talal Bin Abdulaziz Alsaud Institute for Computational Biomedicine, Weill Cornell Medicine, New York, NY, USA. [3]Department of Human Genetics, University of Utah, Salt Lake City, UT, USA. [4]Department of Chemical, Environmental, and Materials Engineering, University of Miami, Coral Gables, FL, USA. [5]Department of Pathology and Laboratory Medicine, University of Miami Miller School of Medicine, Miami, FL, USA. [6]Department of Medicine, University of Miami Miller School of Medicine, Miami, FL, USA. [7]Department of Public Health Sciences, University of Miami Miller School of Medicine, Miami, FL, USA. [8]Department of Molecular & Cellular Pharmacology, University of Miami Miller School of Medicine, Miami, FL, USA. [9]Sylvester Comprehensive Cancer Center, University of Miami Miller School of Medicine, Miami, FL, USA. [10]Institute for Data Science & Computing, University of Miami, Coral Gables, FL, USA. [11]Environmental Health and Safety, University of Miami, Miami, FL, USA. [12]Division of Occupational Health, Safety & Compliance, University of Miami Health System, Miami, FL, USA. [13]Facilities and Operations, University of Miami, Coral Gables, FL, USA. [14]Seed Health, Venice, CA, USA. [15]Harvard Medical School and the Wyss Institute, Boston, MA, USA. [16]The WorldQuant Initiative for Quantitative Prediction, Weill Cornell Medicine, New York, NY, USA. ✉e-mail: btt4001@med.cornell.edu; hmsolo@miami.edu; chm2042@med.cornell.edu

