## [Peer Review File · Nature Communications]

Towards geospatially-resolved public-health surveillance via wastewater sequencingEditorial Note: This manuscript has been previously reviewed at another journal. This document only contains reviewer comments and rebuttal letters for versions considered at *Nature Communications*.

REVIEWERS' COMMENTS

Reviewer #3 (Remarks to the Author):

The authors have worked to improve the manuscript, and most of the earlier comments have been well responded to. I remain of the opinion that aspects of the manuscript overextend on the findings (particularly on diet discussion), but I also appreciate the editors and other reviewers for their continued support. minor comments below.

Line 107/ Line 110: I don't follow the sample size description. On line 109, you state on a "subset of samples", implying that $n = 1272$ is a subset of the total, which earlier you stated "966 (weekly, on average) samples from 34 sites". How many total wastewater samples were collected, how many were sequenced using amplicon sequencing, and how many for metatranscriptions? In the abstract, you state there are 966 amplicon sequences, and 1280 metatranscriptomic samples, are these the same or different samples?

Lines 93-Line 104: The manuscript, as written, does not advance the agenda called for here in the introduction. I agree this is important, I'd suggest the authors move this to the discussion section, specifically because neither their methods nor results link directly to this.

Line 264: Authors state that population dietary choices can be tracked. Can authors analyze their data to track (or rank order or otherwise describe?) population dietary choices in their many catchments from their samples? Evidence supporting from their data would be helpful. Otherwise, suggest authors say "may be useful for".

Line 334: pathogens spelling. Viral names are not italicized or captialized

Reviewer 3, Comment 14:

Thanks for clarification, it is now clear. Suggest "no other wastewater collection site".

Reviewer #3 (Remarks to the Author):

The authors have worked to improve the manuscript, and most of the earlier comments have been well responded to. I remain of the opinion that aspects of the manuscript overextend on the findings (particularly on diet discussion), but I also appreciate the editors and other reviewers for their continued support. minor comments below.

We thank the Reviewer for their effort. In making the recommended changes, we feel the manuscript is again further improved by walking back potentially overextended findings. For example we have moved a paragraph from the introduction (lines 95 to 106 in the tracked version) which was a bit too forward thinking (prior to stating the results) and which we believe can be perceived as an overextension. When we moved this paragraph to the end of the discussion we softened it a bit by substituting "single source" with "primary source" (line 516). In addition we removed sentences at lines 526 and 529 which focused on variations in human relevant events, which we acknowledge can be perceived as overextending the results. We believe that these changes address the reviewer's comment about overextension of the findings. If there are additional areas the Editors feel need to be changed due to overstated claims, we will of course make those changes as needed.

1. Line 107/ Line 110: I don't follow the sample size description. On line 109, you state on a "subset of samples", implying that $n = 1272$ is a subset of the total, which earlier you stated "966 (weekly, on average) samples from 34 sites". How many total wastewater samples were collected, how many were sequenced using amplicon sequencing, and how many for metatranscriptions? In the abstract, you state there are 966 amplicon sequences, and 1280 metatranscriptomic samples, are these the same or different samples?

We agree this was confusing and have updated the text to clarify:

We collected a total of 2,238 samples. 966 were submitted for amplicon sequencing, 1,272 were submitted for metatranscriptomics. The number 1,280 was incorrect and should have read 1,272.

2. Lines 93-Line 104: The manuscript, as written, does not advance the agenda called for here in the introduction. I agree this is important, I'd suggest the authors move this to the discussion section, specifically because neither their methods nor results link directly to this.

We have moved this to the Discussion. The paragraph was slightly modified to fit in the flow of the text, and it (now the last paragraph in the paper) now reads:

"In total, by uniting targeted pathogen tracking with broad monitoring of epidemiologically relevant microbial and environmental signals, there is a potential to use wastewater as a

single source for monitoring multiple dimensions of public health. The development of consistent protocols for sampling, sequencing, and analyzing wastewater data is critical for it to achieve clinical impact. Further, once these protocols are established, robust biomarkers for human population health must be discovered via large-scale association studies, leveraging those that have been successful in other disciplines^{33,37-40}. We claim that this study is a step in this direction: by characterizing the ecology and its variation across time and space, we aimed to lay the groundwork for building a dataset of taxa of interest to be monitored with bulk RNA-sequencing alongside targeted approaches for pathogens of import. Downstream, we propose that a national, or even global, semi-automated surveillance system would serve as a first line of defense against future pandemics, contaminant buildup, ARG tracking, and essential tools to mitigate future public health crises.”

3. Line 264: Authors state that population dietary choices can be tracked. Can authors analyze their data to track (or rank order or otherwise describe?) population dietary choices in their many catchments from their samples? Evidence supporting from their data would be helpful. Otherwise, suggest authors say "may be useful for".

We agree this would be an excellent additional analysis, and we intend to do so in future manuscripts. For the time being, however, we feel additional bioinformatic effort is beyond the scope of this paper. To this end, we have integrated the Reviewer’s suggestion, writing “may be useful for” where recommended.

4. Line 334: pathogens spelling. Viral names are not italicized or capitalized

According to the ICTV (<https://ictv.global/faq/names>), ranks above family should be capitalized but not italicized. Family, genus, and species names should be both capitalized and italicized. We have ensured this passage is in-line with the guidelines, and should the Editors prefer to report these names in a different format, we will of course defer to them.

5. Thanks for clarification, it is now clear. Suggest "no other wastewater collection site".

We have made this change as requested.